# Synthesis and Biological Importance of 2-(thio)ureabenzothiazoles

**DOI:** 10.3390/molecules27186104

**Published:** 2022-09-19

**Authors:** Martha Cecilia Rosales-Hernández, Jessica E. Mendieta-Wejebe, Itzia I. Padilla-Martínez, Efrén V. García-Báez, Alejandro Cruz

**Affiliations:** 1Laboratorio de Biofísica y Biocatálisis, Sección de Estudios de Posgrado e Investigación, Escuela Superior de Medicina, Instituto Politécnico Nacional, Plan de San Luis y Salvador Díaz Mirón s/n, Casco de Santo Tomás, Mexico City 11340, Mexico; 2Instituto Politécnico Nacional-UPIBI, Laboratorio de Química Supramolecular y Nanociencias, Av. Acueducto s/n, Barrio la Laguna Ticomán, Mexico City 07340, Mexico

**Keywords:** 2-aminobenzothiazoles, (thio)ureabenzothiazoles, (thio)phosgene, iso(thio)cyanates, (thio)carbonyldiimidazoles, carbamoyl chlorides, carbon disulfide

## Abstract

The (thio)urea and benzothiazole (BT) derivatives have been shown to have a broad spectrum of biological activities. These groups, when bonded, result in the 2-(thio)ureabenzothizoles (TBT and UBT), which could favor the physicochemical and biological properties. UBTs and TBTs are compounds of great importance in medicinal chemistry. For instance, Frentizole is a UBT derivative used for the treatment of rheumatoid arthritis and systemic lupus erythematosus. The UBTs Bentaluron and Bethabenthiazuron are commercial fungicides used as wood preservatives and herbicides in winter corn crops. On these bases, we prepared this bibliography review, which covers chemical aspects of UBTs and TBTs as potential therapeutic agents as well as their studies on the mechanisms of a variety of pharmacological activities. This work covers synthetic methodologies from 1935 to nowadays, highlighting the most recent approaches to afford UBTs and TBTs with a variety of substituents as illustrated in 42 schemes and 13 figures and concluded with 187 references. In addition, this interesting review is designed on chemical reactions of 2-aminobenzothiazoles (2ABTs) with (thio)phosgenes, iso(thio)cyanates, 1,1′-(thio)carbonyldiimidazoles [(T)CDI]s, (thio)carbamoyl chlorides, and carbon disulfide. This topic will provide information of utility for medicinal chemists dedicated to the design and synthesis of this class of compounds to be tested with respect to their biological activities and be proposed as new pharmacophores.

## 1. Introduction

Studies on urea derivatives as biological modulators of intracellular targets have showed the importance of the urea group that, when incorporated in small molecules, display a broad range of biological activities highlighting the importance of this group in drug development and medicinal chemistry. Many drugs containing the urea group such as Cabozatenib, Sorafenib, and Regorafenib used as anticancer drugs have been developed [1,2]. The urea NH moiety is regarded as a hydrogen bond donor, while its oxygen atom, an excellent acceptor, gives the capability for interacting with a variety of protein targets in several ways to this group. Moreover, the urea moiety favors aqueous solubility due to the strong intermolecular hydrogen bonding formation with different solvents. These interactions play an important role in biological structures and functions, regarding protein and nucleic acid folding, molecular recognition, allostery, signal transduction, and enzymatic catalysis [3].

In general, (un)symmetrical (thio)ureas have been shown to have a broad spectrum of biological and pharmacological activities. These compounds are very interesting due to their use as anti-HIV, antiviral, antimalarial, cytotoxic, anti-inflammatory, antifungal, antimicrobial, antidiabetic, anti-tuberculosis, anti-HCV, and CNS drugs [4,5,6,7,8,9,10,11,12,13,14,15,16,17,18,19,20,21,22,23,24,25,26,27]. Among them, several aryl thiourea derivatives have been used in medicine, industry, and agriculture [28,29,30,31,32]. Moreover, thiourea moiety has been incorporated in many tyrosine kinase inhibitors due to its ability in the formation of hydrogen bonds in the ATP binding cavity of enzymes [33]. For instance, the thiourea derivative YH345A has shown strong protein farnesyl transferase inhibition activity [4]. On the other hand, some heterocyclic thioureas have shown powerful DNA topoisomerase inhibitory activity [10], technological applications such as fluorescence properties [34], corrosion determination of some metals [35], catalysts in chemical reactions [36,37,38,39], and for the extraction of toxic metals [40].

From the drugs used in today’s medicine, many of them possess a heterocyclic nucleus as benzothiazole (BT). Nowadays, BT is a very important phamacophore in medicinal chemistry due to their diversity of pharmacological properties. Developments in biological evaluation of this class of heterocyclic molecules produced changes and promoted the design of new molecules based on their mechanisms of action. BT, one of the family of benzazoles, is an aromatic heterobicyclic compound with a benzene nucleus fused with a thiazole ring. The BT nucleus has been found in marine or terrestrial natural compounds with diverse biological activities. Since 1950, medicinal chemists have been interested in the synthesis of BT derivative compounds, since the pharmacological profile of Riluzole (6-trifluoromethoxy-2-aminobenzothiazole) was found to be a clinically anticonvulsant drug [41,42]. Since then, BT has been an important scaffold with a wide array of interesting biological activities and therapeutic functions. For instance, BT derivatives have been used in the treatment of various diseases such as neurodegenerative disorders, local brain ischemia, central muscle relaxants, and cancer. In the 2000s, few reviews describing the synthetic strategies and biological activities of the BT nucleus were found reported in the literature [43,44,45,46,47]. In the past decade, thirty reviews were found about synthetic methodologies and medicinal activities associated with the BT core as a highly important scaffold for drug development, which has increased rapidly. These reviews were focused on research highlighting anticancer and antimicrobial activities, as well as anticonvulsant, anti-inflammatory, antifungal, antioxidant, antitubercular, antimalarial, antileishmanial, anti-Alzheimer, antitubercular, antidiabetic, and miscellaneous activities [48,49,50,51,52,53,54,55,56,57,58,59,60,61,62,63,64,65,66,67,68,69,70,71,72,73]. Among them, some BT derivatives presented to be useful for the treatment of various diseases including neurodegenerative disorders, local brain ischemia, Huntington’s disease, and cancer. A mini-review about BT derivatives, such as antimicrobial and antiviral [74] and four as anticancer agents, were found [75,76,77,78]. It is worth mentioning that at least ten reviews about this topic were found in the literature from last year until now [79,80,81,82,83,84,85,86,87,88,89]. On the other hand, a review from 2015 to 2020 about the pharmacological activities of BT-related patents was recently reported [90]. Moreover, in 2020, BT derivatives were found to act as multifunctional effectiveness such as antioxidant, sunscreen (filter), antifungal, and anti-proliferative agents [91]. In 2021, we published a literature review on research progress about the condensation of *o*-amino-thiophenoles with carboxylic acids, acid chlorides, amides, nitriles, esters, thioesters, and ortho-esters, including carbon dioxide (CO_2_) as starting materials to access substituted BTs [92].

Nowadays, researchers continuously work on the design and synthesis of BT molecules to obtain more effective derivatives that can be used as drugs [93,94,95,96,97,98,99,100,101,102,103,104]. For instance, scientists are fighting to find drugs against viruses such as influenza and coronavirus. In this sense, two article reviews about the synthesis and structure–activity relationship, as well as various methods to evaluate the antiviral activity against specific viruses of BT derivatives, recently appeared in the literature [105,106].

Since UBTs synthesized by Kauffmann in 1935 were found to be local anesthetics, potent hypoglycemics, and antibacterial agents, these kinds of compounds have had the interest of medicinal chemists [107,108,109,110]. For instance, Frentizole [6-methoxy-phenylureabenzothiazole], Figure 1, a non-toxic UBT drug (IC_50_ = 200 μM), was approved by the FDA for the treatment of rheumatoid arthritis and systemic lupus erythematosus and has been tested for the immunosuppressive and super-immunosuppressive dose levels on the resistance of the mice to viral infections, Figure 1 [111,112]. It was also found that the mean survival time of specific pathogen-free male mice pre-treated with Frentizole or Azathioprine at 100, 50 or 25 mg/kg and infected with herpes simplex and influenza virus was reduced. On the other hand, it is well known that the combination of UBT and TBT derivatives leads to inhibitors of DNA topoisomerase or HIV reverse transcriptase (e.g., **III**, Figure 1) [113,114,115]. In addition, the UBTs Bentaluron and Bethabenthiazuron (Tribunil or Ormet) are commercial fungicides used, respectively, as wood preservative and herbicide in winter corn crops [116,117].

Due to the current relevance of BT derivatives, we focused our attention on (thio)ureabenzothiazole derivatives to prepare a literature review from 1935 to now about the methods of the synthesis of this kind of compound to be analyzed with respect to their biological activity studies. Despite a large number of urease inhibitors being reported and marketed, there is still need of more potent inhibitors with fewer side effects and more efficacy.

In general, the official nomenclature of (thio)ureas is on the base of numbering the *N*1, C2=O(S) and *N*3 of the urea group. Some authors consider this group to be *N*, C=O, *N*’, so in this case, the BT group in (T)BTs is considered to be *N*-benzothiazol-2-yl or *N*1-benzothiazol-2-yl group. The alkyl substituents on *N*′ or *N*3 are represented as *N*′-alkyl or *N*3-alyl. In this work, we consider whether (thio)ureabenzothiazoles can be shortened to (T)UBT. Any substitution on the BT ring and (thio)urea groups was placed on the left of the final name, respectively. For instance, in 6-methyl-methylUBT, the first methyl group is on the 6^th^ position of the BT ring, and the second methyl is on the N3 atom on the urea group. Any di- or tri-substitution on the (thio)urea group can be differentiated by using *N* or *N*1 and *N*′ or *N*3-substitutions.

In general, in this review, 2-aminobenzothiazoles 2ABTs or substituted-2ABTs are starting materials used for the synthesis of TBT or UBT. The introduction of the carbamide group into 2ABTs to obtain UBTs or TBTs were phosgene, isocyanates or isothiocyanates, 1,1′-carbonyldiimidazole (CDI) or 1,1′-thiocarbonyldiimidazole (TCDI), carbamoyl chlorides, and carbon disulfide, as well as some new methodologies, which are commented on. We hope this review will provide information about UBTs and TBTs and their biological activities.

## 2. Results of the Literature Review

In 1935, for the first time, Kaufmann synthesized several substituted UBTs **2**–**3**, bis-6-chloro UBT **4**, substituted phenyl-UBTs **5a**–**e**, and substituted phenyl-TBTs **5f**–**h** from the reaction of the corresponding substituted 2ABTs with urea, cyanic acid, phosgene, benzene-isocyanate, and –thiocyanate in chloroform as solvent, Figure 1 [107].

The same Kauffmann′s methods were used to prepare phenyl-UBTs **6**, substituted TBTs **7** and bis-substituted UBTs **8** in order to examine their antibacterial, acaricidal, and insecticidal activities, toxicity in mice, and relationship between chemical structures and biological actions, Figure 2 [108].

Later, the respective 2ABT or 2-*N*-alkyl-ABT were treated with an alky- or aryl-iso(thio)cyanate or carbamoyl chloride in an aprotic solvent to yield around 81% of the urea compounds **9**, Figure 2 [110]. Refluxing toluene for 4 h was used in the case of substituted-aryl-UBTs and refluxing tetrahydrofuran (THF) for 6 h in the case of substituted-aryl-UBTs. All compounds were tested for immunosuppression in the sheep erythrocyte assay in mice and for antiviral activity against Coxsackie A21 (Coe) virus infection in mice. The most potent immunosuppressant was substituted ureanaphtothiazoles **9a**. One of them was 250 times more active as azathioprine in the sheep erythrocyte test in mice. R^2^ must be an aryl group to be active. For R^1^ = H, the best R^2^ aryl groups were the halogen substituted *p*ClPh, *m*ClPh, and *o*FPh. In the immunosuppressive structure–activity relationship (SAR), the best activity was observed for R groups in the 4^th^ position on the BT ring, for example, 4Cl is even active when R^2^ is cyclohexyl. When R^1^ = Me, the potency decreases to the 50 mg/kg range. In antiviral SAR, the antiviral activity showed to be dependent on the nature of R^2^: if, for example, in R^2^ = 1-naphthyl, adamantyl, and aryl groups, the compounds are quite active.

In 1973, 4,6-disubstituted-β-bromo-propyl-UBTs **10a**–**g** and 4,6-disubstituted-ethenoyl UBTs **11a**–**g** were prepared from the reaction of 4,6-disubstituted-2ABTs with β-bromo-propionyl-isocyanate to be cyclized to produce l-(4,6-disubstituted-benzothiazol-2-yl) dihydrouracils **12**, Figure 3 [118]. A mixture of ether/THF was used for **10a**,**f**,**g** or THF for **10b**–**e**. The reaction mixture refluxed for 2 h. All compounds were shown to have antibacterial, antifungal, and antiprotozoal in vitro effects. The minimal inhibition concentration of 6-thiocyanate-β-bromo-propionyl-UBT **10f** was 50 μg/mL for all tested organisms, which represent a high efficiency.

For instance, Frentizole, 6-methoxy-phenylUBT depicted in Figure 1, was synthetized, and crystals were obtained from ethanol/water to be studied with respect to its structure in the solid state [119].

The reported phenyl-UBT **13** was obtained from the reaction of 2ABT with an excess of phenyl-isocyanate in boiling benzene and recrystallized from EtOH, EtOH-H_2_O, or 1,4-dioxane, Figure 4 [120]. This compound was tested for cytokinin-like activity. The results showed compound **13** inhibited chlorophyll synthesis, perhaps by interfering with the chlorophyll biosynthetic pathway, but not by acting as cytokinin-like. This inhibition was not overcome by the addition of zeatin riboside (ZR).

In 2014, it was reported the first Pd catalyzed C–N coupling of BT-2-ol with phenyl urea via a two-step process involving in situ C–OH bond activation using phosphonium salt as an efficient catalyst system to give out rapid coupling for the synthesis of phenyl-UBT **13** in excellent yield (90%), Figure 4 [121].

A series of 6-substitutedUBTs **14** inhibitors of p56^lck^ were prepared to elucidate their SAR, selectivity, and cell activity in the T-cell proliferation assay, Figure 5 [122]. The urea derivatives **14** were prepared from the corresponding 6-substituted 2ABT when treated with an alkyl- or aryl-isocyanate in dichloromethane and pyridine to form UBTs **14**. Alternatively, the 6-substituted-2ABTs treated with phenyl chloroformate in aqueous THF in the presence of potassium bicarbonate in the form of a phenyl carbamate, which is treated with an amine in THF form the ureas **14**. Compound **14r** was found to be a potent and selective Lck inhibitor with good cellular activity (IC_50_ = 0.004 μM) against T-cell proliferation.

A series of 6-(2,6-dichlorobenzoamidyl)-alkyl-UBTs 15 was prepared from the reaction of *N*-6-(2,6-dichlorobenzoamidyl)-2ABT with the corresponding alkyl-isocyanate in pyridine as a dissolvent at 60 °C for 16 h, Figure 6 [123]. The cytotoxicity text results of all compounds against tumorigenic cell lines were shown. The UBT derivatives **15a**–**e** had the same potency as the amide or urethane derivatives. Selective cytotoxicity against a tumorigenic cell line, WI-38 VA-13 subline 2RA (VA-13), was observed, but not against the normal parental cell line, WI-38. EC_50_ (ng/mL): **15a** = 32, **15b** = 30, **15c** = 28, **15d** = 290, and **15e** = 150.

Several substituted-hepta-O-acetyl-β-D-lactosyl-TBTs **16** were prepared in 64–90% by the condensation of hepta-O-acetyl-β-D-lactosyl-isothiocyanate with substituted-2ABTs, Figure 7 [124]. The structures of these new lactosyl-TBTs have been established on the basis of IR, NMR, and mass spectral studies.

A series of 45 different 6-substituted-aryl-UBTs **17**, **19**, and **20** and 6-substituted-aryl-TBTs **18** were synthesized from the reaction of the corresponding 6-substituted-2ABT with CDI or TCDI in stirring CH_3_CN at room temperature, then reacted with amines in DMF on heating 100 °C, Figure 3 [125]. Using an ELISA-based screening assay, the Frentizole SAR was identified as a novel UBT inhibitor of the interaction of amyloid beta peptide (Aβ) and Aβ-binding alcohol dehydrogenase (ABAD) [Aβ-ABAD], recently implicated in the pathogenesis of Alzheimer’s disease (AD), with a 30-fold improvement in potency. In summary, all synthetized compounds were identified as micro molar inhibitors of the Aβ-ABAD interaction. The compounds **19h** and **19l** showed the most potent inhibition with IC_50_ values of 6.46 and 6.56 μM, respectively.

The phenyl-TBT **21** has been isolated from thermal transformation of 3-phenylamino-5-phenylimino-1,2,4-dithiazole via intramolecular rearrangement through intermediate **A**, Figure 8 [126]. Analogous thermal or acidic catalyzed rearrangements have been described [127].

A series of aryl-UBTs **22** has been synthesized in 40–60% yield from the reaction of the substituted 2ABT with the corresponding aryl-isocyanate in presence of *N*,*N*′-dimethylformamide (DMF), triethylamine (TEA), and dimethylaminopyridine (DMAP) stirring for 8h at room temperature, Figure 9 [128]. The resultant compounds were evaluated for their antiproliferative profiles in human SK-Hep-1 (liver), MDA-MB-231 (breast), and NUGC-3 (gastric) cell lines. Compounds **22e**, **22g**, and **22h** had the potential to moderate inhibitory activities. More of these compounds have been investigated for their ability to inhibit Raf-1 activity. However, they exhibited moderate to poor inhibition.

The 1,1-bistrifluoromethyl-1-alkyl-UBTs **23a**–**c** was prepared by addition of 2ABT to the corresponding bis-fluoroalkyl-isocyanates, Figure 4 [129]. These UBTs were evaluated for their in vitro antiproliferative activities against the human cancer cell lines. The most sensitive cell lines relative to the tested compounds were: **23c** SNB-75 (CNS cancer, log GI_50_ = 5.84) and **24b** UO-31 (renal cancer, log GI_50_ = 5.66), and SR (leukemia, log GI_50_ = 5.44) human cancer cells.

A series of 6-trifluoromethoxy-TBTs **24a**–**f** from 6-trifluoromethoxy-2ABT (Riluzole), a neuroprotective drug in many animal models of brain disease have been synthesized, Figure 5 [130]. The biological activity of synthetized TBTs was preliminarily tested by means of an in vitro protocol of ischemia/reperfusion injury. The results demonstrated that **24a**–**d** significantly attenuated neuronal injury. Selected for the testing of their antioxidant properties, compounds **24a**–**d** were shown to be endowed with a direct reactive oxygen species (ROS) scavenging activity. Compounds **24a** and **24b** were also evaluated for their activity on voltage-dependent Na^+^ and K^+^ currents in neurons from rat piriform cortex. At 50 μM, compound **24b** inhibited the transient Na^+^ current to a much smaller extent than Riluzole, whereas **24d** was almost completely ineffective.

Some 4/6-substituted-phenyl-TBTs **25** were synthesized by the reaction of substituted-2ABT with the corresponding substituted phenyl-isothiocyanate in absolute ethanol, Figure 10 [131]. Compounds **25** were condensed with malonic acid in acetyl chloride to obtain 1-(4/6-substituted-BT-2-yl)-3-phenyl-2-thiobarbituric acid derivatives **26**. All the synthesized compounds were characterized by elemental analysis, infrared (IR), hydrogen nuclear magnetic resonance ^1^H NMR, and Mass spectral studies. Compounds **25** and **26** were screened for their entomological and antibacterial activities. All the synthesized compounds were screened for antibacterial activity at a concentration of 200 μg/mL and 100 μg/mL in DMF using streptomycin and ceftazidime as standard against Gram positive and negative bacteria. Among the synthesized compounds, **26a**, **26f**, and **26g** compounds were found to possess broad-spectrum activities. However, the activities were much less than those of standard antibacterial agents used. These compounds also showed potent antiulcer, anti-inflammatory, and antitumor activities as well as antifeedant activity and acaricidal activity against *Spodoptera litura* and *Tetranychus urticae*, respectively. From the results, these compounds would be better used in drug development to combat bacterial infections and would be better used as antifeedant and acaricidal activities in the future as well.

Firooznia et al., synthesized three 4,7-disubstituted-piperidine UBTs **27a**–**c** from the reaction of the 4,7-disubstituted-2ABT with para nitro phenyl chloroformate, followed by the displacement of *p*-nitro-phenol with an appropriately substituted amine, Figure 11 [132]. The synthesized compounds were evaluated for their selectivity against the A_2A_ and A_1_ receptors. Antagonists of the A_2B_ receptor were used as potential therapeutic agents in models of diabetes, asthma, chronic obstructive pulmonary disease (COPD), and pulmonary fibrosis. Compound **27c** displayed excellent A_2B_ potency, as well as good A_2A_ and A_1_ selectivity, IC_50_ (A_2B_ cAMP) = 20 nm, Ki (A_1_ binding) = 690 nm, and Ki (A_2A_ binding) = 530 nm. A SAR study revealed that compounds with urea derivatives, enhance A_2B_ potency.

The 6-substituted-benzyl UBTs **28a**–**g** were synthetized from the reaction of 6-substituted-2ABT with the in situ generated substituted benzyl-isocyanates obtained from the benzylamine and triphosgene, Figure 12 [133]. To improve yields, *n*-butyl-lithium was used to activate 2ABT (24–70%). In the case of **28g** (R = CN), the 6-tetrazole *p*-methoxybenzyl-UBT **29g** was obtained in 91% using sodium azide NaN_3_ in microwave radiation. All compounds were evaluated for their inhibition of glycogen synthease kinase-3 (GSK-3) activity. Several compounds were identified to reduce in vitro GSK-3β activity beneath 50% at a concentration of 10 μM. The SAR of the library justified the synthesis of the UBT **29g** (IC_50_ = 140 nM), which displayed more than twofold enhanced activity compared with the reference compound 1-(4-methoxybenzyl)-3-(5-nitrothiazol-2-yl)urea (AR-A014418: IC_50_ = 330 nM).

A series of sixteen differentially substituted-(phenyl)-methyl-phosphonate-TBTs **30a**–**p** were synthetized from substituted 2ABTs and O,O′-di-alkyl-isothiocyanate-(phenyl)methyl-phosphonates in heating dry CH_3_CN at 90 °C in 30 min under microwave irradiation with a power input of 120 W, Figure 13 [134]. The products were obtained in good to excellent yields (40–81%), shorter reaction times, milder reaction conditions, and simple purification procedures. The compounds possessed broad-spectrum insecticidal and antiviral activities against *Tobacco Mosaic Virus* (TMV) in vivo. Two compounds, **30a** (R^1^ = 6F, R = *^n^*Pr) and **31l** (R^1^ = OMe; R = *^n^*Bu), had remarkably high in vitro insecticidal activities against *Plutella xylostella* compared with the control insecticide Avermectin. Furthermore, all were associated with moderate to good anti-TMV activities.

Caputo et al. synthesized two sets of 6-substituted Aryl-UBTs **31a**–**e** and **32a**–**e** by reacting substituted 2ABTs with aryl-isocyanates in dry dichloromethane at room temperature (**31a**–**e**, 60–85% and **32a**–**e**, 81–93%) and evaluated in an in vitro primary anticancer assay against a panel of 60 human tumor cell lines, Figure 14 [135]. Compounds **32a** and **32c** showed good anticancer activities, more marked for compound **32c**. All compounds in a preliminary in vitro assay as inhibitors of the ubiquitin-activating enzyme (E1) lacked significant activity. The UBT scaffold **32c** showed considerable growth inhibitory activities against different human tumor cell lines such as leukemia (log GI_50_ value −5.93), non-small cell lung (log GI_50_ value −6.0), colon cancer (log GI_50_ value −5.89), CNS cancer (log GI_50_ value −5.73), melanoma (log GI_50_ value −5.89), ovarian cancer (log GI_50_ value −5.74), renal cancer (log GI_50_ value −5.90), prostate cancer (log GI_50_ value −5.72), and breast cancer (log GI_50_ value −6.0) as compared with the reference drug 5-fluorouracil NSC 19893.

The appropriate 4/6-arylsubstituted-2ABT and the respective aryl-isothiocyanate was reacted in a mixture of DMAP (5 mol %) in DMF to furnish 6/4-arylsubstituted-TBT derivatives **33a**–**l** in 83–91% yield, Figure 6 [115]. All compounds were evaluated for cytotoxic activity against two human monocytic cell lines (U 937, THP-1) and a mouse melanoma cell line (B16-F10). Based on their IC_50_ values, almost all compounds had significant antiproliferative activity on U 937 and B16-F10 cells, with **33b**, **33e**, **33f**, **33k**, **33c**, and **33h** being the most actives. Compound **33e** demonstrated to have the best antiproliferative activity against the U-937 cell line. The IC_50_ values of compound **33e** were higher (16.23 ± 0.81 μM), (47.73 ± 2.39 μM), and (34.58 ± 1.73 μM) as compared to standard compound etoposide IC_50_ values (17.94 ± 0.89 μM), (18.69 ± 0.94 μM), and (2.16 ± 0.11 μM)) against U-937, B16-F10 and THP-1 cell lines, respectively.

Azam and coworkers synthetized aryl-UBT derivatives **34a**–**s** from the reaction of 2ABT in acetonitrile as dissolvent with the dropwise addition of the appropriate aryl-isocyanate stirred at room temperature until the completion of the reaction (1–4 h), Figure 15 [136]. All compounds were tested as anti-Parkinsonian agents with an improved pharmacological profile in haloperidol-induced catalepsy and oxidative stress in mice. All compounds were active in alleviating haloperidol-induced catalepsyin mice. Furfuryl **34i** and 2-methoxy **34c** emerged as potent agents. With exception of 2-chloro, 5-trifluoromethyl-substituted analog **34q**, and halogen substituted derivatives **34n**–**p** exhibited moderate antiparkinsonian activity. Molecular docking studies of these compounds with adenosine A2A receptor exhibited very good binding interactions and warrants further studies to confirm their binding with human A2A receptor for the design and development of better therapy for Parkinson′s disease PD.

The 6-methoxy-*m*-methoxyUBTs **35a**, 6-methoxy-*m*-methoxyTBTs **35b**, and their hydrolyzed compounds **36a**,**b** were synthesized via amide coupling starting from the reaction of 6-methoxy-2ABT with *m*-methoxy-benzo-isocyanate or *m*-methoxy-benzo-isothiocyanate in pyridine at 100 °C for 20 h, respectively, Figure 16 [137]. The ether cleavage of the methoxy groups was carried out with boron tribromide (BBr_3_) in methylene chloride (CH_2_Cl_2_) at −278 °C to room temperature to yield compounds **36a** and **36b**. The three-unit bridge resulted in the inactive urea **36a** against human 17β-hydroxysteroid dehydrogenase 1 (17β-HSD1), but the more lipophilic thiourea **36b** showed a moderate inhibitory activity (70b: 62% inhibition at 1 µM).

Patents from 2007 to 2012 reported the reaction of 6-bromo-5-iodo-2ABT or 7-bromo,5-iodo-2ABT with ethyl-isocyanate in 1,4-dioxane at 80°C for 12 h under N_2_ to afford the corresponding 1-(5.6- or 1-(5,7-dihalogen)ethyl-UBTs, Figure 7 [138,139,140,141,142]. In addition, one hundred and thirty-eight 1-(4-fluoro-5-bromo-7-substituted-ethylUBTs were obtained from the reaction of the corresponding dihalogenide-2ABT with methyl(chloro)thioformate, phenyl-chloroformate, or *p*-nitrophenyl chloroformate in the presence of a base as pyridine or TEA in a solvent as dichloromethane, chloroform, carbon tetrachloride or mixtures, Figure 7. The resultant 1-(5,6- or 1-(5,7-dihalogenide-BT-2-yl)-3-ethylurea in DMF/H_2_O was treated with pyridine-3-boronic acid and potassium phosphate (K_3_PO_4_) to generate the corresponding 6-fluoro-5-substituted-ethylUBT **37**, 5,7-disubstituted-ethylUBTs compounds **38**–**41**, and 4-fluoro-5,7-disubstituted-ethylUBT as **42** in approximately 65% yield.

Ethyl-UBTs **37**–**42** were tested against Gram-positive and Gram-negative pathogens. The minimum inhibitory concentration (MIC_90_) values for **37** were found to be: *S*. *aureus* = 0.06 μg/mL, *S*. *pneumoniae* = 0.015 μg/mL, *S*. *epidermidis* = 0.03 μg/mL, and *E*. *faecalis* = 0.25 μg/mL.

Compound **38** exhibited important inhibitory activity against *S*. *aureus* GyrB (IC_50_ 0.014 μg/mL) and strong antibacterial activity against VR *E*. *faecalis*, VR *E*. *faecium*, FQR *S*. *pneumoniae*, *S*. *aureus*, and *M*. *catarrhalis* with a MIC value of less than 25 μg/mL. The solubility and antimicrobial profiles of **39** were improved at physiological pH from 6.25 μg/mL to 50 μg/mL by bonding a carboxylic acid on pyridine group (**39**), while maintaining antibacterial activity against target pathogens (*S*. *aureus* GyrB IC_50_ 0.001 μg/mL, *E*. *faecalis* MIC 0.12 μg/mL). In addition, benzothiazole ethyl ureas **40** and **41** were synthesized by changing the carboxylate in compound **38** by a cyclic amine [78,79]. Both compounds inhibited the turnover of ATP by both the DNA gyrase and topoisomerase IV enzymes, with mean IC_50s_ ranging from 0.0033 μg/mL to 0.046 μg/mL. These compounds were also tested against six major Gram-positive pathogens, exhibiting potent inhibition with MIC_90_ values from 0.015 μg/mL for *S*. *pneumoniae* to 0.25 μg/mL for *S*. *aureus*, compared with the MIC_90_ values of 2 μg/mL and 4 μg/mL, respectively, for linezolid and 16 μg/mL and 16 μg/mL, respectively, for levofloxacin. The tested compounds were also potent against *S*. *aureus* drug-resistant strains, including VRE, MRSA, VRSA, linezolid-non-susceptible *S*. *aureus*, daptomycin-non-susceptible, methicillin-resistant *S*. *epidermidis*, penicillin resistant *S*. *pneumoniae*, and FQ-*S*. *pneumoniae* strains. Moreover, no cross-resistance was observed caused by the mechanisms responsible for conferring resistance to these antibiotics. Both compounds also showed a promising pharmacokinetic profile in rats and mice [78].

Two additional patents described the synthesis of 4-fluoro-5,7-disubstitutedbenzothiazole ureas as compound **42** [75]. All compounds were tested for their DNA supercoiling assays for GyrB inhibition and a DNA relaxation assay for ParE inhibition. Under both assay conditions, few compounds showed activity in the sub-micromolar range (IC_50_ values 30–550 nM) and were found to be active in antibacterial assays against Gram-positive pathogens.

The same procedure was used to synthetize a series of 6-substituted-ethylUBTs **43**, **44** and 6-substituted-R^1^ethylUBTs **45** as effective inhibitors against wild-type (wt) and T_315_I mutant Bcr-Abl Kinases, Figure 17 [143]. Through a structure-based drug design, Nocodazole was modified, varying the C2 and C6 groups. It was found that the introduction of polar groups at the terminal ethyl group enhanced physicochemical properties and potency in cellular inhibition. Several compounds inhibited the kinase activity of both wild-type Bcr-Abl and the T_315_I mutant with IC_50_ values (**43a**, IC_50_(wt) = 0.06 nM and IC_50_(T_315_I) = 0.11 nM) and exhibited good antiproliferative effects on Ba/F_3_ cell lines transformed with either wild-type or T_315_I mutant Bcr-Abl.

By using ethyl isocyanate in 1,4-dioxane at 80 °C overnight, the 5-OBn and 7-bromo-2ABT was transformed to the 5,7-disubstituted-ethylUBT **46** in 75% yield, subsequent pyridine-2-yl group was coupled to the C-7 to afford compound **47** in 70% yield, Figure 18 [144]. Compound **47** was Bn-deprotected (97%), converted to the triflate (68%), and a Miyaura borylation reaction yielded 88% of the key borinic acid intermediate **48** for further derivatization through a Suzuki cross-coupling of the corresponding 5-bromo pyrimidine with **49**, followed by saponification to yield 5,7-disubstituted-ethylUBTs **49a**–**d**, **50a**–**h**, and **51a**–**d**, Figure 18. The synthetized compounds were tested as bacterial DNA gyrase and topoisomerase IV inhibitors. Antibacterial properties were demonstrated by activity against DNA gyrase ATPase and potent activity against *Staphylococcus aureus*, *Enterococcus faecalis*, *Streptococcus pyogenes*, and *Haemophilus influenzae*. Compounds **50a** and **50b** had IC_50_ values for S. aureus topoisomerase IV (0.012 and 0.008 μg/mL, respectively) comparable to their S. aureus DNA gyrase ATPase IC_50_ values. These compounds also showed specificity for bacterial topoisomerases, with no inhibition of human topoisomerase II. The compounds **49a**–**h** bearing a α-substituent to the carboxylic acid group, resulting in excellent solubility and favorable pharmacokinetic properties.

Compounds **50a** and **50b** in Figure 18 were evaluated for their biochemical, antibacterial, and pharmacokinetic properties [145]. Both compounds inhibited the ATPase activity of GyrB and ParE with IC_50_ of < 0.1 μg/mL. Prevention of DNA supercoiling by DNA gyrase was observed. These compounds inhibited the growth of a range of bacterial organisms, including *staphylococci*, *streptococci*, *enterococci*, Clostridium difficile, and selected Gram-negative respiratory pathogens. MIC_90_s against clinical isolates ranged from 0.015 μg/mL for *Streptococcus pneumoniae* to 0.25 μg/mL for *Staphylococcus aureus*. No cross-resistance with common drug-resistance phenotypes was observed. In addition, no synergistic or antagonistic interactions between compound **50a** or **50b** and other antibiotics, as the topoisomerase inhibitors novobiocin and levofloxacin, were detected in checkerboard experiments. The frequencies of spontaneous resistance for *S. aureus* were < 2.3 × 10^−10^ with compound **50a** and < 5.8 × 10^−11^ with compound **50b** at concentrations to 8 folds the MICs. These values indicate a multi-targeting mechanism of action. The pharmacokinetic properties of both compounds were profiled in rats. Following intravenous administration, compound **50b** showed an approximately 3-fold improvement over compound **50a** in terms of both the clearance and the area under the concentration–time curve. The measured oral bioavailability of compound **50b** was 47.7%.

Palmer et al. used the same procedure to obtain a series of 5-substituted alcohol-containing ethyl UBTs **52a**–**s**, **53a**–**g**, and **54a**–**k**, which were identified to have superior antibacterial activity and drug-like properties, Figure 8 [146]. The 5-substituted alcohol-containing ethyl-UBTs **52a**–**s** were prepared to improve the pharmacokinetic profile. The SAR study of these series revealed that these compounds had potent antibacterial activity against a primary panel of pathogenic bacteria compared to the carboxylic acid **50c**, Figure 18. Compounds **53a**–**c** displayed potent antibacterial activity against *S. aureus* 29213 (MIC 0.03–0.06 μg/mL), *S. pyogenes* (MIC 0.06–0.12 μg/mL), and *H. influenza* 49247a (MIC 0.25–1 μg/mL). Compounds **52a**–**c** also displayed significant inhibitory activity against purified *S. aureus* T173 GyrB (IC_50_ 0.25 μg/mL all cases) and *S. pyogenes* ParE (IC_50_ to 0.5–1 μg/mL). Compounds **52a**, **d**–**m** were investigated for the scope and limitations of the alcohol-containing substituents. In general, compounds showed potency in the nanomolar range against DNA gyrase ATPase in a malachite green assay. The introduction of bulky groups at the tertiary, pseudo benzylic position of the pyrimidine ring resulted in a remarkable loss of activity, particularly against the ParE mutant strain of *S. pyogenes* (**52d** diethyl, IC_50_ 8 μg/mL; **52f** cyclohexyl ring system, IC_50_ > 16 μg/mL). However, increasing on-target ATPase inhibition was observed when a heteroatom was introduced into the alicyclic 6-membered ring (**52g**–**j**), IC_50s_ *S. aureus* GryB 0.25–8; *S. pyogenes* ParE 1–8 μg/mL), suggesting a conformational role, possibly a flattening of the ring. Reduced antibacterial activity against *S. aureus* 29213 (MIC > 16 μg/mL), *S. pyogenes* 51339 (MIC 8 μg/mL; IC_50_ > 16 μg/mL), and *H*. *influenza* 49247a (MIC 16 μg/mL) was observed when the ring was contracted to an azetidine (**52k**) and leaving the free NH group more exposed. However, strong potency (MIC 0.06–0.5 μg/mL) across the entire tested bacterial stains (with IC_50s_ values of 0.5 and 2 μg/mL against *S. aureus* GryB and *S. pyogenes* ParE, respectively) was observed in the case of physically constrained oxetane analogue **53m**.

The secondary alcohols **52n**–**q**, **s** and the diol-containing molecule **52r** were explored to enhance the solubility. The extension of R^1^ to a longer chain alkyl and small cycloalkyl groups **52n**, **52o**, **52q** exhibited no effect on the potency of the series toward the enzyme. However, the large *t*-butyl moiety **52p** showed a substantial drop in activity against *Haemophilus influenza*. Moreover, a remarkable drop in activity against *S*. *aureus* in the presence of serum was observed. The morpholine group in **52s** and diol in **52r**, designed to enhance hydrophilicity, was well tolerated. Further, the extension of the diol-containing series resulted in compounds **53b**–**f**. In this series, the introduction of small groups did not hamper antibacterial activity significantly. However, diol **53d** showed a relative intolerance of the large rings. Compounds **54a**–**f** enhanced hydrophilicity and reduced protein binding resulted in potent Gram-positive antibacterial activity. Further, introduction of alcohol-containing moieties at the C5 position of the BT core and modifications at the C7 position yielded compounds **54a**–**k**. These compounds with enhanced hydrophilicity and reduced protein binding resulted in potent Gram-positive antibacterial activity. Compounds **54f**–**h** containing 5-membered rings substituted with ether or alcohol containing moieties, enhanced the unbound fraction from 9% to 19%. However, this enhancement was partially compensated for by a moderate increase in MIC. The substitution of azetidine moieties in compounds **54i**–**k** enhanced this property further.

In a typical reaction, *ortho*-, *meta*-, or *para*-tolyl-isocyanates were treated with 2ABT in the presence of 1,4-dioxane at room temperature to synthetize three tolyl-UBTs **55a**–**c** to evaluate their inhibitory effects on α-chymotrypsin enzyme, Figure 9 [147]. It was found the *para*-substituted *N*-(1,3-benzothiazol-2-yl)-*N*′-(4-methylphenyl) urea **55c** derivative substantially inhibited α-chymotrypsin activity (IC_50_ = 20.6 ± 0.06 μM). Due to the presences of the bulky 1,3-benzothiazol-2-yl group at one nitrogen of the urea bridge, the activity is enhanced by the reduction of the steric hindrance in the order *ortho* > *metha* > *para*.

The 6-bromo-2ABT was reacted successively with CDI and substituted ethylamine to yield 6-bromo ethyl-UBTs **56** in 74–90%, Figure 19 [148]. Intermediates **56** were reacted successively with bis(pinacolato)diboron and intermediate **57**, catalyzed by PdCl_2_(dppf), to yield a series of 6-(2,3-disubstituted pyridine-5-yl)-ethyl-substituted-UBTs **58a**–**l** in 33–51%. The antiproliferative activities of the synthesized compounds in vitro were evaluated against HCT116, MCF-7 U87MG, and A549 cell lines. The compounds with potent antiproliferative activity were tested for their acute oral toxicity and inhibitory activity against PI3Ks and mTORC1. The results indicate that the compounds with R^1^ = 2-dialkylaminoethyl moiety **58b**, **58f**, **58k**, and **58l** retain the antiproliferative activity and inhibitory activity against PI3K and mTOR. In addition, their acute oral toxicity reduced dramatically. Moreover, compound **58f** can effectively inhibit tumor growth in a mouse S180 homograft model. These findings suggest that derivatives **58f**, **58k**, and **58l** can serve as potent PI3K inhibitors and anticancer agents with low toxicity.

Using the same procedure, compounds **59m**–**q** were synthetized in a 40–45% yield to compare the activities with different substituents at the 3-position of pyridine ring, Figure 10. In the case of pyridine ring **59m**, the activity dramatically dropped.

The 6-sulfonamide-2ABT was reacted with one equivalent of carbon disulfide and one or two equivalents of dimethyl sulfate in the presence of one equivalent or two equivalents of sodium hydroxide to afford compounds **60** in 54% and **62** in 67%, Figure 20. Furthermore, compound **60** was reacted with ethylenediamine or piperazine in dioxane at room temperature to give the bis-TBTs **61a** or **61b**, in 21 and 20.7%, respectively. Under the same reaction conditions, compound **62** produced the bis-Methyl-iso-TBTs **63a** or **63b** in 14 and 17%, respectively [149]. All compounds were investigated as inhibitors of the four isoforms of the metalloenzyme carbonic anhydrase (CA, EC 4.2.1.1), the cytosolic CA I and II, and the tumor-associated isozymes CA IX and XII. Docking studies showed favorable interactions between the scaffolds of these new inhibitors and the active sites of the investigated CA isoforms. Compounds **61a**,**b** and **63a**,**b** acted as highly potent inhibitors of the tumor-associated hCA IX and hCA XII with K_I_s in the nanomolar range (2.1–2.6 and 4.3–4.8 nM), compared to the standard drugs **AAZ** (25 and 5.8 nM) and **EZA** (34 and 22 nM). The cytosolic isozyme hCA II was also inhibited with K_I_s ranging from 3.5 to 4.7 nM, compared with the standard drugs (12 and 8 nM). On the other hand, from molecular docking studies, compounds **61a**,**b** and **63a**,**b** were found as potent inhibitors against the slow cytosolic isoform hCA I with K_Is_ in the range of 12.0–37.4 nM, compared to the standard drugs (250 and 25 nM).

A series of seventeen different picoline-amide based UBT derivatives **64a**, **64b**, **65**, and **66a**–**n**, were synthesized as shown in Figure 21 [150]. The treatment of 2ABT with the appropriate aliphatic isocyanate in dioxane afforded compounds **64a** and **64b** in approximately 87% yield, while the thiourea derivative **65** was obtained in only a 10% yield from the reaction of 2ABT with ethyl-iso-thiocyanate in pyridine. On the other hand, the target compounds **66a**–**n** were prepared via a two-step one-pot reaction. The 2ABT was first converted into its isocyanate intermediate by the treatment of CDI in DMF and followed by the reaction with suitable aliphatic amine. All compounds were tested for their inhibitory activities against the wild-type ABL at 10 μM and showed strong enzymatic inhibition, 93.3–100%. Their IC_50_ values were found to be 18.2–285 nM. Compounds **64a**, **65**, **66a**, **66f**, **66g**, **66i**, **66l**, and **66n** were tested over the mutant type ABLT315I and displayed promising potency with IC_50_ values of 39.9–511 nM.

The 3,3,3-trifluoroethyl-isocyanate was used for the synthesis of a series of substituted tri-fluoroethyl-UBTs **67a**–**d** in 70–95% yield, Figure 22 [151]. These compounds were tested at a single high dose (10^−5^ M). The moderate anticancer activity against some types of cancer on the individual human cell lines for leukemia, non-small cell lung cancer and renal cancer were shown. All compounds showed anticancer activity on individual cell lines. Activity was notable on individual cell lines against leukemia, non-small cell lung cancer, and renal cancer. The COMPARE algorithm analysis revealed that possible mechanisms of action of these ureas include alkylating agent **67c**, DNA antimetabolites **67b**–**d**, RNA/DNA antimetabolites **67d**, topoisomerase II inhibitor **67b**, and antimitotic agent **67d**.

In **2016**, a three-component reaction of *o*-amino-thiophenol disulfide, copper cyanide, and aryl cyanates were used for the synthesis of *N*,*N*-dimethyl-UBT **68** and phenyl-UBTs **69**, Figure 23 [152]. This transformation is based on an oxidative copper-mediated S-cyanation as a key step, then a cyclization sequence enabling a rapid and efficient synthesis of 2ABT, which reacts with the corresponding electrophile to produce thiourea **68** in 37% or urea **69** in 55%.

A series of substituted *p*-benzene-sulfonamide-TBTs **70a**–**c** were synthesized from substituted 2ABTs and *p*-isothiocyanate-benzene-sulfonamide as starting materials, Figure 24 [153]. The structures of the compounds were established from elemental analyses, IR, ^1^H-NMR, ^13^C-NMR, and mass spectral data analysis. All compounds were evaluated for their in vitro anticancer activity against various cancer cell lines. Compound **70a** exhibited good activity higher than or comparable to the reference drugs, 2′7′dichlorofluorescein (DCF) and Doxorubicin, except the breast cancer line. As a trial to suggest the mechanism of action of the active compounds, the molecular docking on the active site of the mitogen kinase enzyme (MK-2) was performed, and the results showed compound **70a** may represent a good candidate for further biological investigations as anticancer agents.

A series of 6-substituted-alkyl(aryl)-UBTs and 6-substituted-alkyl(aryl)-TBTs **71a**–**v** were synthesized from 6-substituted-2ABTs and alkyl/cycloalkyl/aryl-isocyanates or isothiocyanates in refluxing toluene, Figure 25 [154]. The reaction rates and yields depended on substituents introduced on the BT ring and the isocyanates and isothiocyanates. The reaction rate was greater with an electron-donating group on the BT ring such as ethyl C_2_H_5_ or methyl CH_3_. The rate was very low with an electron-withdrawing group, such as nitrogen dioxide NO_2_. Isocyanates were more reactive than isothiocyanates due to the greater electronegativity of the oxygen atom compared to that of the sulfur atom. Compounds were tested as myorelaxants and inhibitors of insulin secretion. Compounds **71u** and **71v** showed high myorelaxant activity. The 6-substituted-alkyl-UBTs **71f**, **71g**, and **71t**–**v** containing strong electron-withdrawing groups as NO_2_, CN at the 6-position, and an alkyl group linked to the urea or the thiourea function were found to be the most potent compounds. Some compounds were tested on rat pancreatic islets provoked a marked inhibition of insulin secretion, among which **71a** exhibited a clear tissue selectivity for pancreatic β-cells.

A series of UBTs **72a**–**m** tethered with a pyridyl-amide moiety by ether linkage at the 6-position of BT was synthesized as potent anticancer sorafenib analogs, Figure 26 [155]. The synthesis was achieved by treating the 6-substituted 2ABT with the corresponding aryl isocyanate in either anhydrous DMF or acetonitrile under argon atmosphere for compounds **72a**–**j**. Compounds **72k**–**m** were prepared via a two-step one-pot reaction. The 6-substituted 2ABT was converted into the isocyanate by treatment with CDI in DMF, followed by the reaction with the proper aromatic amine. Twelve of these derivatives were analyzed for its antiproliferative activity over a panel of 60 human cancer cell lines at a single-dose concentration of 10 μM. Compounds **72a**–**d** were more potent than the sorafenib used in the treatment of renal cell carcinoma, and their GI_50_ values were determined. Compound **72b** showed good inhibitory activities against ACHN (renal cancer cells lines) and A-498 (human kidney carcinoma cell line) with GI_50_ values of 0.542 µM and 1.02 µM, respectively. This compound also showed efficacy against UO-31 and RXF 393 cell lines. Compounds **72a** and **72d** exhibited excellent antiproliferative activities with low IG_50_ values of 1.85 and 2.10 µM against RCC and ACHN cell lines, respectively. The SAR study revealed that sorafenib analogues possess anti proliferative activity due to the presence of both urea spacer and phenyl di-substitution. Compound **72b** demonstrated the highest CLogP value was the most lipophilic and potent derivative in the low µM range.

To the in situ generated 2-isocyanateBT from substituted 2ABT and CDI in DMF at room temperature, the appropriate aniline was added, and the reaction mixture was heated at 100 °C for 3 h to generate the corresponding UBTs **73a** and **73b**, Figure 27 [156]. The in vitro anticancer evaluation of **73b** showed substantial broad-spectrum antiproliferative activity against 60 human cancer cell lines. It showed GI_50_ values of 51.4 and 19 nM against leukemia K-562 and colon carcinoma KM12 cell lines, respectively. The kinase screening of compound **73b** revealed its nanomolar-level inhibitory activity of certain oncogenic kinases implicated in both tumorigenesis and angiogenesis. Compound **73b** displays IC_50_ values of 0.82, 3.81, and 53 nM toward Tie2, TrkA, and ABL-1(wild-type and T315I mutant) kinases, respectively. Moreover, **73b** is orally bioavailable with a favorable in vivo pharmacokinetic profile. Compound **73b** may serve as a promising candidate for the further development of potent anticancer chemotherapeutics.

Molecular modeling study was performed to evaluate the recognition of a TBT **74**, and its Cu(II), Co(II), and Ni(II) metal complexes were synthesized, Figure 28 [157]. They were characterized by micro analysis, IR, ^1^H-NMR, EPR, UV-Visible spectral analyses, molar conductance, and thermal analysis studies. These studies revealed that the metal complexes have distorted octahedral geometry. The recognition of target compounds at the 3MNG binding pocket was evaluated by molecular modeling. The copper complex selectively binds to the crucial amino acid residues in the active site of 3MNG. The in vitro antioxidant activity of the ligand and its metal complexes was assayed by radical scavenging activity (DPPH, H_2_O_2_ and NO) and ferric-reducing antioxidant power (FRAP) methods. The ligand showed moderate antioxidant activity, whereas the metal complexes exhibited better antioxidant activity than that of the ligand, with the copper complex being the most potent antioxidant.

Selected 6-subtituted-2ABTs were activated with one equivalent of CDI to obtain their corresponding intermediates **B** in excellent yields (90–95%), which were treated with a substituted aromatic amine to produce two series of 6-substituted-aryl-UBTs **75a**–**zt** in moderate-to-excellent yields (36–99%) along with novel insights into the structure and activity relationships for the inhibition of amyloid-beta binding alcohol dehydrogenase (ABAD), Figure 29 [158]. Compounds **75zg** and **75zi** showed potent ABAD inhibition, whereas compound **75zi** exhibited comparable cytotoxicity with the Frentizole standard; however, this was a one-fold higher cytotoxicity than the parent Riluzole standard. The calculated and experimental physical–chemical properties of the most potent compounds showed promising features for blood–brain barrier penetration, BBB.

On the other hand, compounds **75a**–**zb** in Figure 29 were evaluated for their inhibitory activity on CK1 and their potential to cross the BBB was predicted using the central nervous system–multiparameter optimization (CNS-MPO) model and eventually parallel artificial membrane permeability assay (PAMPA) [159]. Several compounds were found to be sub-micromolar CK1 inhibitors, identified using compound **75q** as being the best hit (IC_50_ = 0.16 mM/1.92 mM for CK1_δ_ resp. CK1_ε_). However, compounds **75e** and **75s** were shown to be low micromolar inhibitors of both CK1 and ABAD, and hence they present a potential novel class of dual-acting anti-AD therapeutics. The results of PAMPA for **75e** and **75s** suggest that the compounds should be able to penetrate into the brain.

A series of 6-substituted aryl-(thio)urea-BTs and GBTs **76a**–**o** was synthetized, Figure 30 [160]. The synthesis started with the activation of the corresponding 6-substituted 2ABT using CDI in refluxing DMF or TCDI in refluxing acetonitrile (MeCN), Figure 30. In the next step, the reactive imidazolyl intermediate **C** was treated with the corresponding aniline in DMF at 60 °C to obtain the 6-substituted thio(urea) products **76**. Guanidine analogues were prepared by treating the corresponding thiourea with mercury oxide in methanolic ammonia solution. Compound **76l** was identified as the most promising hit compound with good inhibitory activity (IC_50_ = 3.06 ± 0.40 µM) and an acceptable cytotoxicity profile comparable to the Frentizole. The acceptable physical–chemical properties of the guanidine compound **76l** suggest its capability to permeate through the BBB, making compound **76l** a novel lead structure for further development and biological assessment.

The 5-substituted-TBTs **77a**–**d** were prepared in excellent yields (60–68%) by the reaction of 5-substituted 2ABTs with carbon disulfide (CS_2_) and dimethyl sulfate followed by ammonolysis of the intermediate **D**, Figure 31 [161]. The cytotoxic activity was screened for antitumor activity against human breast cancer cells (MCF-7), human cervix epithelial carcinoma (HeLa), a human colon cancer cell line (HT-29), a human leukemia cell line (K-562), and a mouse neuroblastoma cell line (Neuro-2a) using cisplatin as a reference by MTT assay. The results provide experimental evidence that the compounds induce apoptosis in cancer cell lines. According to flow cytometry results, the treatment of HT-29 cells with **77d** produced a large population of apoptotic cell (79.45%), which was 1.2-fold higher than that produced by cisplatin (65.28%) at the same concentration.

Fluoroquinolone-TBT derivatives **78a**–**n** were synthetized from a mixture of 6-substituted 2ABT, fluoroquinolone, and CS_2_ in basic medium (NaHCO_3_/DMF) and refluxed for 10 h, Figure 32 [162]. All compounds were tested against bacterial strains. Some compounds exhibited excellent antibacterial activity against *Staphylococcus auerus*, *Escherichia coli*, *Bacillus subtilis*, and *Pseudomonas aeruginosa* bacterial strains. Among all compounds, the 6-nitro substituted UBT, along with norfloxacin **78b** and gatifloxacin **78l**, showed MIC_50_ μg/mL when tested against *S*. *auerus*. Moreover, compounds **78d**, **78f**, and **78l** showed superior MIC (15, 10, and 15 μg/mL, respectively) against *B*. *subtilis.* The results reveal that the compounds have significant antibacterial potential and are suitable candidates for further exploration.

Fifteen differentially substituted-TBTs **79a**–**o** were synthetized from the reaction of commercially available 2ABT and the corresponding alkyl- or aryl-isothiocyanate in acetonitrile, Figure 33 [163]. Substituted-TBTs **79a**–**g** were cyclized to 1,3-thiazolidin-4-ones **80a**–**g**. Molecular structure of compounds **79d**, **79m**, **79o**, and **80c** was determined by X-ray crystallography. All compounds were evaluated on their cytotoxicity against human leukemia/lymphoma- and solid tumor-derived cell lines and of their antiviral activity against HIV-1 and representatives of ssRNA and dsDNA viruses. Compound **80e** showed activity against the HIV-1 wild type and against variants carrying clinically relevant mutations. A colorimetric enzyme immunoassay clarified its mode of action as a non-nucleoside inhibitor of the reverse transcriptase.

A mixture of 6-methoxy-2ABT, BMZ derivatives, CS_2_, basic medium (NaHCO_3_) and DMF was refluxed for 10 h. to afford a series of *N*-(6-methoxybenzo[*d*]thia-zol-2-yl)-2-substituted phenyl-1*H*-benz[*d*]imidazole-1-carbothioamides **81a**–**h**, Figure 34 [164]. All compounds were characterized and screened for their in vitro antimicrobial activity against selected bacterial and fungal strains. These compounds were also evaluated for their antimalarial activity against *P*. *falciparum*. Antimicrobial activity screening results showed compounds **81b** against *P*. *aeruginosa*, **81c** against *S*. *aureus* and *E*. *coli*, **81d** against *E*. *coli* and *P*. *aeruginosa*, and **81g** against *E*. *coli* emerged as prospective antibacterial leads with excellent activity (MIC 12.5–62.5 μg/mL). Only one fungal strain, *C*. *albicans*, was susceptible towards synthesized compounds. On the other hand, compounds **81c** and **81h** exhibited antimalarial activity with IC_50_ values of 0.18 and 0.11 μg/mL as compared to standard drugs chloroquine (IC_50_ 0.020 μg/mL) and quinine (IC_50_ 0.268 μg/mL).

Compound AHS-211 was modified to prove the role of the urea linker to preserve the bioactive conformation and led to the development of 6-ydroxy-arylUBTs **84a**–**c** as promising selective Dyrk1A inhibitors, Figure 35 [165]. The compounds **84a**–**c** were synthetized through the reaction of the 6-methoxy-2ABT with phenyl-isocyanates in DMF at room temperature to generate the 6-methoxy-arylUBTs **82a**–**c**, Figure 35, or alternatively through the reaction of 6-methoxy-2ABT with phenyl-chloroformate in the presence of pyridine as a base and dioxane as solvent to give the carbamate as intermediate, which was further in situ reacted with amine derivatives in refluxing dioxane to give the 6-methoxy-ethylsubstitutedUBTs **83a** and **83b**, Figure 35. UBT **83b** was hydrolyzed with aqueous KOH solution in THF at room temperature for 14 h, then the carboxylic acid of **82c** was reacted with the corresponding amine and HBTU at room temperature overnight to generate 6-methoxy-arylUBTs **85a**,**b**. Compounds **82a** and **85a**,**b** were ether dealkylated in the presence of Phosphorous tribromide (PBr_3__)_ in CH_2_Cl_2_ as a dissolvent at room temperature for 20 h to generate compounds **84a**–**c**. Although the extension with the 4-benzyl amide in compound **84b** enhanced the potency toward Dyrk1A (IC_50_ = 0.063 μM) by more than 2-fold compared with **84a** and AHS-211, it did not show any improvement in selectivity over Dyrk1B. On the other hand, compound **83c** was almost similar to AHS-211 in potency against Dyrk1A, yet it showed more than a 15-fold selectivity for Dyrk1A over Dyrk1B. Importantly, both **84b** and **84c** displayed superior selectivity for Dyrk1A over Dyrk2 when compared with AHS-211. Additionally, **84c** displayed good selectivity for Dyrk1A over Clk1, one of the most common off targets of many reported Dyrk1A inhibitors, with a selectivity factor > 7 (IC_50_ of **84c** against Clk1 = 730 nM, [ATP] = 15 μM). The design concept of attaching a urea linker to the BT core, followed by a scaffold expansion, led to a new class of potent and selective Dyrk1A inhibitors. The best compound, **84c**, did not inhibit the homologous Dyrk2 isoform and even showed a remarkable 15-fold selectivity over the most closely related isoform Dyrk1B. In addition, the activity toward the atypical kinase haspin that was strongly inhibited by most of the previously reported Dyrk1A inhibitors was abolished with **84c**. Together with the favorable balance of lipophilic/hydrophilic functions, further testing of **84c** as a potential agent for the treatment of Dyrk1A-related neuropathological disorders could be envisaged.

From the 6-aminoethyl-UBT obtained from the reaction of 6-nitro-2ABT with ethyl isocyanate in refluxing toluene 24 h in the presence of TEA, followed by reduction of the nitro group to amine, Figure 36 [166]. A series of ethyl-urea derivatives of 6-amino-2ABT, named **86a**–**I**, were synthetized and evaluated for *E. coli* DNA gyrase inhibition using an in vitro DNA gyrase supercoiling assay. The most potent DNA gyrase inhibitors were 6-substituted UBTs **86c**–**e**, **86g**, and **86h** with IC_50_ values in the low micromolar range. The most promising inhibitors identified were evaluated against selected Gram-positive and Gram-negative bacterial strains. Compound **86d** showed a MIC of 50μ M against an *E. coli* efflux pump-defectives train, which suggests that the efflux decreases the on-target concentrations of these compounds.

Three series of 6-substituted-aryl UBTs **87a**–**l**, **88a**–**g**, and **89a**–**d** were designed and synthetized, Figure 11 [167]. The synthesis proceeded according to the two-step reaction process; 6-methoxy-2ABT reacted with CDI to give the mono-substituted intermediate. This intermediate was subsequently reacted with the respective substituted aniline to give a final 6-substituted-aryl UBTs **87a**–**l**. To obtain compounds **87f**, **87g**, and **88i**, the N-Boc protective group was cleaved under acidic conditions. Compound **90a** was prepared from the 2(N-Me-amino)-BT with the corresponding 3-chloro, 4-methoxy-phenylisocyanate in THF at room temperature. All compounds were evaluated for in vitro 17β-HSD10 inhibitory ability. Compounds **87d** and **88c** showed the most promising 17β-HSD10 inhibitory activity in enzymatic assays, although the orthogonal screens indicated that **88c** could be inhibiting 17β-HSD10 in an unfavorable manner. Key structure–activity relationships were established and validated with a urea linker, and a 4-phenolic moiety with a 3-halogen substitution confirmed to be essential for compound 17β-HSD10 inhibitory ability. Furthermore, a bulky 6-substitution (e.g., *t*-butyl) on BT appeared to be the most promising, potentially occupying the chemical space more effectively within the binding site. Positively, the most promising compounds were also shown to have an inhibitory effect at a cellular level with limited cytotoxicity, and all hit compounds displayed a more favorable kinetic mechanism of action (reversible mixed inhibition) to other previously published work. These findings provided significant structural activity insight into the 17β-HSD10 inhibitor compound design and were the most promising observations to date. With further hit optimizing and neuronal cellular evaluation to determine if these compounds are protective against Aβ-mediated cytotoxicity, this could potentially lead to a novel class of therapeutics for AD.

The 2,3-cyclization reaction of unsymmetrical 6-substituted-TBTs **F** with 2-bromoacetophenone in the presence of TEA afforded *N*-(6-substitued-1,3-benzothiazol-2-yl)-4-phenyl-1,3-thiazol-2(3H)-imine compounds **90a**–**d**, Figure 37 [168]. Also, 2-Me-carbamodithioate-BTs **E** were reacted with methyl anthranilate to give the in situ thiourea intermediate **G**, which was cyclized to afford 3-(benzo[d]thiazol-2-yl)-2-thioxo-2,3-dihydroquinazolin-4(1H)-one derivatives **90a**–**d**. Compounds **90a**–**d** were evaluated for anti-tumor activity against MCF-7, MDA-MD-231, HT-29, HeLa, Neuro-**90a**, K-562, and L-929 cell lines, and the results from the MTT-assay revealed the best cytotoxicity for **90b** compound (IC_50_ = 5.94 ± 1.98 μM).

The 2-dithiomethylcarboimidateBT **92**, derived from 2ABT, was reacted with *l*-glycine, *l*-alanine, *l*-phenyl-glycine, *l*-phenyl-alanine, *l*-valine, and *l*-leucine to afford a series the isolable sodium salts of the SMe-iso-thiourea carboxylates **93a**–**f**, whose hydrolysis of the SMe group and methylation of the isolable UBT-carboxylates with methyl iodide in stirring DMF as solvent affords the UBTs methyl esters **94a**–**f**, Figure 38 [169]. The structures of synthesized compounds were established by ^1^H and ^13^C NMR, and the structures of methyl esters SMe-iso-thiourea-BTs **93a**–**f** were derived from (*l*)-glycine, (*l*)-alanine, (*l*)-phenyl-glycine, and (*l*)-leucine by X-ray diffraction analysis.

A library of ten 6-substituted-*p*-methoxybenzylUBTs **95a**–**j** was designed and synthetized, Figure 39 [170]. Molecular modeling studies with the proposed covalent inhibitors were performed using the Maestro Schrödinger Drug Discovery Suite. The biological activities of the compounds were also tested.

The in situ generated ammonium hydrochloride salt of *p*-methoxy-benzyl-amine was reacted with CDI for 7 min to yield the intermediate *N*-(4-methoxybenzyl) carbamoyl-imidazole **H** in a quantitative yield, which was coupled with the corresponding 6-substituted 2ABT to form 6-substituted-*p*-methoxybenzylUBTs **95a**–**j** and to improve the literature yields of **95a** and **95h**, which were increased from 70 and 26% to 99% and quantitative yield, respectively.

Compounds **95a**–**c**, **95b**,**c**, **95f**,**g**, and **95j**, were evaluated for GSK-3β inhibitory activity using a luminescent assay. The potent GSK-3β inhibitor AR-A014418 (IC_50_ = 0.072 ± 0.043) was used as reference standard.

The inhibitory ability of its BT-equivalent, **95a** (**28g**, Figure 12), is known in the literature.

The HMK inhibitor **95c** was more potent than the corresponding acetyl-derivative, **95b**, confirming the structure–activity influence of the halomethyl-ketone moiety to increase inhibitory activity. Furthermore, the biological activities of the vinyl and ethyl ketones **95f** and **95g** were low, whereas the acrylamide **95j** displayed insignificant inhibition.

A set of 60 substituted-aryl-UBTs **96a**–**zzc** and the 6Cl-(3Cl,4OHphenyl)-TBT **97** was prepared from the corresponding substituted-2ABT when treated with CDI or TCDI, followed by the addition of the corresponding substituted aniline, Figure 12 [171]. The compounds were evaluated for their inhibitory ability and mechanism of action against the human 17β-HSD10. The most potent inhibitors contained 3-chloro and 4-hydroxy substitution on the phenyl ring moiety. Among these, compounds **96zx**, **96zzc** exhibited IC_50_ values of 1–2 µM and showed an uncompetitive mechanism of action with respect to aceto-acetyl-CoA. These uncompetitive UBTs inhibitors of 17β-HSD10 were considered promising compounds as potential drugs for neurodegenerative diseases.

Several Frentizole derivatives **98a**–**k** were selected by molecular docking as potential nicotinate mononucleotide adenylyl-transferase (NadD) inhibitors for antimicrobial activity evaluation, Figure 13 [172]. Compounds **98f** and **98h**–**k** showed antimicrobial activity against Gram-positive bacteria-*Staphylococcus Aureus*, Methicillin-resistant *Staphylococcus aureus* (except for **98f** and **98j**), *Staphylococcus epidermidis* (except for **98j**), and Vancomycin-resistant *Enterococcus* (only **98k**). However, even the best obtained MICs and MBCs were substantially higher than values corresponding to standard benzalkonium bromide (BAC_14_). Therefore, none of the tested derivatives can be considered to be a novel promising antimicrobial agent.

Chiral UBT **99a** and chiral (T)UBTs **99b**,**c** were synthesized from the reaction of 2ABT with sodium hydride NaH in DMF, followed by addition of the respective chiral-(1-isocyanatoethyl)benzene, and stirred for 12 h at room temperature, Figure 40 [173]. All compounds were in vitro evaluated for their antimicrobial activity against *B*. *cereus, S*. *aureus, E*. *coli*, and *P*. *aeruginosa*. The results indicated all compounds were only active in Gram-positive bacteria. The UBT **99a** and TBTs **99b** and **99c** with either *R*- or *S*-configurations or had no antibacterial properties.

As depicted in Figure 41, forty nine combined substituted-aryl-UBTs **100a**–**p** that were synthetized from substituted-2ABTs and substituted-phenyl-isocyanates in ketone by an easy and cheap synthetic route, Figure 41 [174]. Synthetized compounds were studied on their antibacterial activity against Gram-positive and Gram-negative strains. Compounds **100f** and **100j** showed the highest antimicrobial activity against *Staphylococcus aureus*; they were more active than triclocarban (TCC), with MIC values of 8 µg/mL versus 16 µg/mL of TCC. Moreover, compound **100c** was much more active than TCC against *Enterococcus faecalis*, a Gram-positive bacterium that is strongly responsible for nosocomial infections. Finally, compound **100p**, even though less active than the others, exerted an interesting bactericidal action.

An iodine-catalyzed reaction of 6-substituted-2ABTs and isocyanides for the synthesis of 6-substituted-substituted UBTs **101a**–**i** (48–60%) via a metal-free isocyanide insertion reaction was reported, Figure 42 [175]. Introducing a simple method for the synthesis of desired UBT skeletons and the use of more acceptable iodine molecules instead of expensive transition metal catalysts are the most important advantages of this strategy.

It is important to mention that several substituted-substituted TBTs had been used since 1968 as intermediates to synthetize 2-guanidinebenzothiazoles (GBTs), for example, substituted-substituted-TBTs **102** were obtained as isolable intermediates from the reaction of substituted-2ABTs with alkyl- or aryl-thiocyanates followed by the amination-desulfurization reaction with PbO, HgCl_2_, HgO, CuSO_4_.5H_2_O, CH_3_I to afford GBTs **103**, Figure 43 [176,177,178,179,180,181,182,183,184].

Also, interesting 6-substituted-sugar TBTs **104a**–**c** were obtained in 56–77% yields from the reaction of the respective 6-substituted-2ABT and the corresponding per-O-acetylated isothiocyanate lactose derivative in dry pyridine, which were transformed to the respective GBT derivative, Figure 14 [185].

## 3. Conclusions

From the literature review, benzothiazole, was found to be a bioactive and structurally simple benzofuzed heterocyclic compound that plays an important role in medicinal chemistry. It was observed that, depending on the functional group present on the BT molecule, the group plays an important role in the physicochemical properties. On the quest to discover better medicinal agents, researchers should understand the relative contributions of each functional group on the BT ring. It can become a part of the development and discovery of new drugs with potential biological activity. During the last decade, efforts have been taken to synthesize medicinally important BT derivatives, and, among them, researchers have discovered many (T)UBT derivatives showing promising biological activities.

In the present review, efforts were taken to summarize the different methodologies used for the synthesis of (T)UBT derivatives along with their biological activity. In general, 2ABTs were reacted with (alkyl-)aryl-iso(thio)cyanates, 1′-(thio)carbonyldiimidazole (T)CDI, carbamoyl chlorides, and carbon disulfide for the synthesis of (T)UBTs. It is hoped that this review will benefit budding researchers in the field of benzothiazole urea-based drug designing.

This topic is interesting, because it will provide information of utility for medicinal chemists dedicated to the design and synthesis of this class of compounds to be tested with respect to their biological activities and be proposed as new pharmacophores.

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
