# Peer review of "Synthesis and Biological Importance of 2-(thio)ureabenzothiazoles"

_molecules, 2022, doi:10.3390/molecules27186104_

Round 1
Reviewer 1 Report
In this review, the authors focused on different methodologies used for the synthesis of pharmacophore 2-(thio)ureabenzothizole and their derivatives along with their biological activity against different diseases including rheumatoid arthritis, systemic lupus erythematosus, anticancer, antibacterial, antimalarials etc.
1) This review will allow design and the development of benzothizole based therapeutics against different diseases and infections.
2) Abstract and introduction are clear and easy to understand. Overall well-structured review.
3) Sufficient information about the previous study findings is presented for readers to follow the present study and procedures.
I believe this review is suitable for publication in Molecules
Author Response
Thanks a lot for the comments and suggestions
Reviewer 2 Report
The paper entitled “Synthesis and Biological Importance of 2-(thio)ureabenzothiazoles”, covers synthetic methodologies from 1935 to nowadays, highlighting the most recent approaches to afford UBTs and TBTs with a variety of substituents. This work could provide information of utility for medicinal chemists dedicated to the design and synthesis of this class of compounds to be tested with respect to their biological activities. I would like to recommend it for accepted after revision noted below:
1) Please polish the molecular structures in this manuscript. Authors should note the bond length and bond angle.
2) Line 271 in page 8: “2IO-4CO2Me” Whether it is wrong? And “4OH-5CO2Me” should not superscript, please check carefully for the whole paper.
3) There are many hyperlinks in this paper. What is the purpose of the authors? For example Line 865: antibacterial activity.
4) Line 239 in page 7: “N-6-(2,6-dichlorobenzoamidyl)-2ABT” and Line 1023 in page 32: “the N-(4-methoxybenzyl) carbamoyl-imidazole”. The “N-” should use the same format. Please check carefully for the whole paper.
5) Line 411 in page 12: “N,N-dimethyl formamide (DMF)” Abbreviations should first appear at the front of the paper.
6) The names of some journals are not abbreviated. Such as: Line 1215 in page 37: “Molecular Diversity” should be “Mol. Divers.”. Lines 1311 in page 39: “Chinese J. Org. Chem.” should be “Chin. J. Org. Chem.”.
Author Response
Reviewer 2
Response to Comments and Suggestions for Authors
The authors tanks reviewer the comments and suggestions
The authors re-reviewed our article and taking in account these suggestions, were restructured. Right away we response the comments and suggestions
1.- Please polish the molecular structures in this manuscript. Authors should note the bond length and bond angle.
R = All structures in figures and schemes were adjusted with respect to their bonds and angles
2) Line 271 in page 8: “2IO-4CO2Me” Whether it is wrong? And “4OH-5CO2Me” should not superscript, please check carefully for the whole paper.
R = It was corrected to 2I-4CO2Me and 4OH-5CO2Me and was revised for the whole paper
3) There are many hyperlinks in this paper. What is the purpose of the authors? For example Line 865: antibacterial activity.
R = The hyperlinks were eliminated
4) Line 239 in page 7: “N-6-(2,6-dichlorobenzoamidyl)-2ABT” and Line 1023 in page 32: “the N-(4-methoxybenzyl) carbamoyl-imidazole”. The “N-” should use the same format. Please check carefully for the whole paper.
R = The N-substituted compounds were homogeinized as N-substituted for the whole paper.
5) Line 411 in page 12: “N,N-dimethyl formamide (DMF)” Abbreviations should first appear at the front of the paper.
R = Complete name of abbreviations only appear in parenthesis at the first time in the text.
6) The names of some journals are not abbreviated. Such as: Line 1215 in page 37: “Molecular Diversity” should be “Mol. Divers.”. Lines 1311 in page 39: “Chinese J. Org. Chem.” should be “Chin. J. Org. Chem.”.
R = It was corrected in the bbliographyc section
Reviewer 3 Report
The authors have organized several literature reports on the synthesis and biological importance of 2-(thio)-ureabenzothiazoles. However, the review article is difficult to read and understand. There are several missed edits and typos. In the present format, this review’s importance is unclear. This review lacks uniformity in presentation (sometime listing yield, sometimes not for example). The authors appear to not have carefully crosschecked the numbers of the compounds or the numbers of the figures, and sometimes appear to not have carefully read the original report/literature.
1. Some of the schemes has yield for the reactions and most of them has not, please maintain uniformity throughout the review and yield to the schemes.
2. Figure representation do not justify the reaction sequence in a synthesis review, please replace the figures with proper schemes.
3. Reactants in some of the schemes, reactant or starting material has number but most of them are missing, please maintain uniformity throughout the review.
4. There are a numbers of arbitrary underlined words throughout the manuscript. Are there any reasons behind that!!
5. Page 2, lines 72-75; “In the last past decade …………. increased rapidly” is an incomplete sentence.
6. Page 2, lines 75-80; Please provide individual reference for each individual activity property separately (ref 48-78).
7. Page 4, line 138; “Results and discussion’ a review doesn’t conduct any experiments so there should not be a ‘results and discussion’ section. Please provide an alternative heading for this section.
8. Page 4, line 161; “….. yield 81 of the urea…” what is this ‘81’?
9. Page 5, Figure 2. What are R1and R2?
10. Page 5, paragraph 1, Hard to understand anything without proper numbering of the discussed compounds.
11. Page 6, Paragraph 1, lines 195-206; What is this compound 13? Where is its structure? Why is this discussion about Frentizole crystal structure in the middle of the systemic review about synthetic and biological importance of these compounds!!
12. Page 6, line 230-232; “Compound 15r…………. proliferation” but the original report describes compound ‘15v’ has the similar IC50 value like 16r (0.004 uM).
13. Page 7, Scheme 7; the per-O-acetylated lactosyl isothiocyanate is number as 2 and 2-amino benzothiazole derivatives as 5 but both are wrong, they have already used these numbers for different compounds.
14. Page 8, line 260; “……. of 45 6-substituted…” or “…. of 45 different 6-substituted…..”
15. Page 8, Paragraph 1, lines 268, 269; “The compounds 20h and 20l …………… respectively” The fact is that compound 20h and 20l was found most potent (IC50 6.46 uM and 6.56 uM respectively) in that series but octanol-water partition coefficients (log P) values are not the parameters for that. This octanol-water partition coefficients (log P) values were determined to check their ability to cross blood-brain barrier. More disturbingly the authors copied and paste this portion of the sentence as the original manuscript where they have used reference number ‘8’ after log P value of 1.15. Original paper “octanol–water partition coefficients (log P) of 1.34 and 1.15,8 respectively” and here they wrote “octanol-water partition coefficients (log P) of 1.34 and 1.15,8 respectively”
16. Page 8, figure 3; Some of the structures have -OH some of them has -HO, please maintain uniformity. What is 2IO group (compound 20n)
17. Page 8, line 227; “………intermediate B”, Where is ‘A’ in the manuscript!
18. Page 11, line 367; “…….. reference compound,….” What reference compound?
19. Page 11, line 376; “….sixteen substituted…” or “…..sixteen ‘differentially’ substituted…..”!!
20. Page 13, scheme 15; No numbering for the compounds but the describing text has numbers.
21. Page 14, line 459; “…..hand, 138 1-(4-fluoro-5……………..” what is this 138?
22. Page 14, line 463; “…. Figure 7.79” Is this ‘79’ a reference here? If yes, then correct the format. But reference number 79 in the manuscript represents a review article from Chinese J. Org. Chem. Then why using this review reference in the description of a certain reaction protocol.
23. Page 18, line 612; Not figure 8 but figure 9.
24. Page 19, line 643, not figure 9 but figure 10.
25. Figure number 9 repeated twice in the review. Check figure numbers there after.
26. Page 21, line 673; “…. 17 picolinamide…” or “…… 17 different picolinamide…..”!
27. Page 21, line 677; “…. From compound 2 via…” From compound 2??? Doesn’t look like that
28. Page 26, line 833; “….. MeCN, scheme 1.” Scheme 1 doesn’t contain this reaction sequence with DMF.
29. Page 26, scheme 30; What are b1,2 and c in the reaction sequence? No numbering for the compounds!!!
30. Page 27, line 878; “Fifteen substituted…” or “Fifteen differentially substituted…..”
31. Page 35, scheme 43; Compound numbers 12 and 13 are wrong.
32. Page 35, line 1122 and page 36, figure 13; Compound number 17 is wrong. Per-O-acetylated lactose derivative was 17 and this one 6-subtituted a-methyl glucoside and they are completely different.
33. Page 39, line 1333, provide proper abbreviated name for the journal.
34. For many references last page numbers are missing.
and so on …..
Author Response
Reviewer 3
Response to Comments and Suggestions for Authors
The authors tanks reviewer the comments and suggestions
The authors re-reviewed our article and taking in account these suggestions, were restructured. Right away we response the comments and suggestions
1.- Some of the schemes has yield for the reactions and most of them has not, please maintain uniformity throughout the review and yield to the schemes.
1.- R = The yields were included in the schemes. Some yields were not found in the respective cite
2.- Figure representation do not justify the reaction sequence in a synthesis review, please replace the figures with proper schemes.
2.- R = so has not to be repetitive, we used figures in well known methods. For instance, in the use of cyanates or thicyanates.
3.- Reactants in some of the schemes, reactant or starting material has number but most of them are missing, please maintain uniformity throughout the review.
3.- R = The same of the suggestion 2, to avoid repetition, reactants well known were not numbered, however they are mentioned in the text.
4.- There are a numbers of arbitrary underlined words throughout the manuscript. Are there any reasons behind that!!
4.- no, any reason, they were eliminated
5.- Page 2, lines 72-75; “In the last past decade …………. increased rapidly” is an incomplete sentence.
5.- R = The sentence was corrected to be a complete sentence
6.- Page 2, lines 75-80; Please provide individual reference for each individual activity property separately (ref 48-78).
6.- R = It was difficult because these references are article reviews. However some were separated.
7.- Page 4, line 138; “Results and discussion’ a review doesn’t conduct any experiments so there should not be a ‘results and discussion’ section. Please provide an alternative heading for this section.
7.- R = This was change to Results of the literature review
8.- Page 4, line 161; “….. yield 81 of the urea…” what is this ‘81’?
8.- R = It was corrected (81% yield)
9.- Page 5, Figure 2. What are R1and R2?
9.- R = The reference not show all the R, R1 and R2 groups, however, in the text, some R, R1 and R2 groups are commented.
10.- Page 5, paragraph 1, Hard to understand anything without proper numbering of the discussed compounds.
10.- R = In this reference the discussion is general and only some importance of the presence of the R, R1 and R2 groups in the biological activity were commented.
11.- Page 6, Paragraph 1, lines 195-206; What is this compound 13? Where is its structure? Why is this discussion about Frentizole crystal structure in the middle of the systemic review about synthetic and biological importance of these compounds!!
11.- R = Frentizole structure is represented in the figure 1 and was not numbered. The text was renumbered without compound 13.
12.- Page 6, line 230-232; “Compound 15r…………. proliferation” but the original report describes compound ‘15v’ has the similar IC50 value like 16r (0.004 uM).
12.- R = we were wrong, R = compound 15r, now 14r has a IC50 = 1.1 mM proliferation and 15v, now 14v IC50 = 2.8 mM, it was corrected
13.- Page 7, Scheme 7; the per-O-acetylated lactosyl isothiocyanate is number as 2 and 2-amino benzothiazole derivatives as 5 but both are wrong, they have already used these numbers for different compounds.
13.- R = The numbers were wrong and were eliminated because of the isocyanate ad substituted-2ABTcompounds are easily identified.
14.- Page 8, line 260; “……. of 45 6-substituted…” or “…. of 45 different 6-substituted…..”
14.- R = It was changed for “forty five different”
15.- Page 8, Paragraph 1, lines 268, 269; “The compounds 20h and 20l …………… respectively” The fact is that compound 20h and 20l was found most potent (IC50 6.46 uM and 6.56 uM respectively) in that series but octanol-water partition coefficients (log P) values are not the parameters for that. This octanol-water partition coefficients (log P) values were determined to check their ability to cross blood-brain barrier. More disturbingly the authors copied and paste this portion of the sentence as the original manuscript where they have used reference number ‘8’ after log P value of 1.15. Original paper “octanol–water partition coefficients (log P) of 1.34 and 1.15,8 respectively” and here they wrote “octanol-water partition coefficients (log P) of 1.34 and 1.15,8 respectively”
15.- R = We were wrong, the parameters are the IC50 values. It was corrected including the values
16.- Page 8, figure 3; Some of the structures have -OH some of them has -HO, please maintain uniformity. What is 2IO group (compound 20n)
16.- R = All structures have -OH, 2IO is wrong, it was changed for 2I.
- Page 8, line 227; “………intermediate B”, Where is ‘A’ in the manuscript!
17.- R = as changed by A and subsequently the intermediates corrected in all text.
- Page 11, line 367; “…….. reference compound,….” What reference compound?
18.- R = Reference compound is 1-(4-methoxybenzyl)-3-(5-nitrothiazol-2-yl)urea (AR-A014418)
- Page 11, line 376; “….sixteen substituted…” or “…..sixteen ‘differentially’ substituted…..”!!
19.- R= sixteen differentially
- Page 13, scheme 15; No numbering for the compounds but the describing text has numbers.
20.- R = letters of compounds are sequential (2MePh is 34a and so on, until 4NO2Ph is 34s)
- Page 14, line 459; “…..hand, 138 1-(4-fluoro-5……………..” what is this 138?
21.- R = 138 was substituted by one hundred and thirty eight
- Page 14, line 463; “…. Figure 7.79” Is this ‘79’ a reference here? If yes, then correct the format. But reference number 79 in the manuscript represents a review article from Chinese J. Org. Chem. Then why using this review reference in the description of a certain reaction protocol.
22.- R = This reference was wrong and was eliminated. These protocols were found in addition in the patent references [140-144]
- Page 18, line 612; Not figure 8 but figure 9.
23.- R = It was corrected
- Page 19, line 643, not figure 9 but figure 10.
24.- R = It was corrected in all text.
- Figure number 9 repeated twice in the review. Check figure numbers there after.
25.- R = It was checked
- Page 21, line 673; “…. 17 picolinamide…” or “…… 17 different picolinamide…..”!
26.- R = It was corrected to seventeen different picolinamide.
- Page 21, line 677; “…. From compound 2 via…” From compound 2??? Doesn’t look like that
27.- R = It was corrected
- Page 26, line 833; “….. MeCN, scheme 1.” Scheme 1 doesn’t contain this reaction sequence with DMF.
28.- R = It was corrected
- Page 26, scheme 30; What are b1,2 and c in the reaction sequence? No numbering for the compounds!!!
29.- R = b1, b2 and c are reactives and were included in the scheme 30.
- Page 27, line 878; “Fifteen substituted…” or “Fifteen differentially substituted…..”
30.- R = Ok “fifteen differentially
- Page 35, scheme 43; Compound numbers 12 and 13 are wrong.
31.- R = It was corrected
- Page 35, line 1122 and page 36, figure 13; Compound number 17 is wrong. Per-O-acetylated lactose derivative was 17 and this one 6-subtituted a-methyl glucoside and they are completely different.
32.- R = This was corrected
- Page 39, line 1333, provide proper abbreviated name for the journal.
33.- R = Proper abbreviated name for the journal were provided.
- For many references last page numbers are missing.
34.- R = Some journals as Molecules, J. Mol Strct., Antibiotics, Int. J. Mol. Sci. etc. gives the article number instead of page numbers.
Round 2
Reviewer 3 Report
The current format is in much better shape and can be accepted.
Author Response
Response to suggestions of the reviewer (red colour)
1) Please polish the molecular structures in this manuscript. Authors should note the bond length and bond angle.
The molecular structures were polished in all the text considering the bond lenght and the bond angles.
2) Line 271 in page 8: “2IO-4CO2Me” Whether it is wrong? And “4OH-5CO2Me” should not superscript, please check carefully for the whole paper.
2IO-4CO2Me was changed to 2I-4CO2Me (I = iodo). These groups are in position 2 and position 4 on the phenyl ring in compound 19.
4OH-5CO2Me was changed to 4OH-5CO2Me. The two groups are in position 4 and 5 of the phenyl ring in compound 19.
3) There are many hyperlinks in this paper. What is the purpose of the authors? For example Line 865: antibacterial activity.
All the hiperlinks in the text were eliminated
4) Line 239 in page 7: “N-6-(2,6-dichlorobenzoamidyl)-2ABT” and Line 1023 in page 32: “the N-(4-methoxybenzyl) carbamoyl-imidazole”. The “N-” should use the same format. Please check carefully for the whole paper.
We use the italic format N for substituents on nitrogen atoms in the whole paper
5) Line 411 in page 12: “N,N-dimethyl formamide (DMF)” Abbreviations should first appear at the front of the paper.
We think the name with their abbreviations sould be at the first time when it appear in the text, then only the abbreviations can be used.
6) The names of some journals are not abbreviated. Such as: Line 1215 in page 37: “Molecular Diversity” should be “Mol. Divers.”. Lines 1311 in page 39: “Chinese J. Org. Chem.” should be “Chin. J. Org. Chem.”
All the names of the journals were reviewed and some were corrected.
Thanks a lot for review our article
